# Genetic Signature of River Capture Imprinted in *Schizopygopsis* Fish from the Eastern Tibetan Plateau

**DOI:** 10.3390/genes15091148

**Published:** 2024-08-31

**Authors:** Lijun He, Yonghong Bi, David Weese, Jie Wu, Shasha Xu, Huimin Ren, Fenfen Zhang, Xueqing Liu, Lei Chen, Jing Zhang

**Affiliations:** 1State Key Laboratory of Estuarine and Coastal Research, East China Normal University, Shanghai 200241, China; 51183904038@stu.ecnu.edu.cn (S.X.); 51183904026@stu.ecnu.edu.cn (H.R.); ffzhang@sklec.ecnu.edu.cn (F.Z.); jzhang@sklec.ecnu.edu.cn (J.Z.); 2Hubei Key Laboratory of Three Gorges Project for Conservation of Fishes, Chinese Sturgeon Research Institute, China Three Gorges Corporation, Yichang 443100, China; liu_xueqing@ctg.com.cn; 3State Key Laboratory of Freshwater Ecology and Biotechnology, Institute of Hydrobiology, Chinese Academy of Sciences, Wuhan 430072, China; biyh@ihb.ac.cn; 4Department of Biological and Environmental Sciences, Georgia College & State University, Milledgeville, GA 31061, USA; david.weese@gcsu.edu; 5Shanghai Entomological Museum, Center for Excellence in Molecular Plant Sciences, Chinese Academy of Sciences, Shanghai 200032, China; jiewu@cemps.ac.cn

**Keywords:** first bend, *Schizopygopsis*, phylogeograpy, phylogeny, river evolution, upper Changjiang/Yangtze River

## Abstract

Some East Asian rivers experienced repeated rearrangements due to Indian–Asian Plates’ collisions and an uplift of the Tibetan Plateau. For the upper Changjiang (Yangtze/Jinsha River), its ancient south-flowing course and subsequent capture by the middle Changjiang at the First Bend (FB) remained controversial. The DNA of freshwater fishes possess novel evolutionary signals of these tectonic events. In this study, mtDNA *Cyt b* sequences of endemic *Schizopygopsis* fish belonging to a highly specialized grade of the Schizothoracinae from the eastern Tibetan Plateau were used to infer the palaeo-drainages connectivity history of the upper Changjiang system. Through phylogenetic reconstruction, a new clade D of *Schizopygopsis* with three genetic clusters and subclusters (DI, DII, DIIIa, and DIIIb) were identified from the upper Yalong, Changjiang, and Yellow Rivers; the Shuiluo River; the FB-upper Changjiang; and the Litang River; respectively. Ancient drainage connections and capture signals were indicated based on these cladogenesis events and ancestral origin inference: (1) the upper Yalong River likely acted as a dispersal origin of *Schizopygopsis* fish to the adjacent upper Yellow and Changjiang Rivers at ca. 0.34 Ma; (2) the Litang River seemed to have directly drained into the upper Changjiang/Yangtze/Jinsha River before its capture by the Yalong River at ca. 0.90 Ma; (3) the Shuiluo River likely flowed south along a course parallel to the upper Changjiang before their connection through Hutiao Gorge; (4) a palaeo-lake across the contemporary Shuiluo, Litang, and Yalong Rivers was inferred to have served as an ancestral origin of clade D of *Schizopygopsis* at 1.56 Ma. Therefore, this study sheds light on disentangling ambiguous palaeo-drainage history through integrating biological and geological evidence.

## 1. Introduction

The collision of the Indian and Asian plates during the Miocene promoted the uplifting of the Qinghai–Tibetan Plateau and reversed the topographic tilt of East Asia from west tilting to east tilting [1]. Some Asian rivers were subsequently reorganized through river capture and reversal events [2]. For example, the Oligocene palaeo-Changjiang/Yangtze River was composed of three main segments with different courses [3,4,5,6,7,8] (Figure 1): the lower Changjiang/Yangtze River flowed eastward, draining into the Yellow Sea or East China Sea; the middle Changjiang/Yangtze River/palaeo-Chuanjiang River flowed westward; and the upper Changjiang/Yangtze River/Jinsha River likely flowed southward along the eastern margin of the Tibetan Plateau and emptied into the South China Sea through the Red River [9]. Subsequently, the middle Yangzte River and upper Yangtze River successively experienced capture/reversal through redirecting to the east into the lower Yangtze River due to the Tibetan Plateau’s abrupt uplifts [2] (Appendix A). As a tributary of the upper Changjiang/Yangtze River, the palaeo-Shuiluo River is thought to have flowed south through the Hutiao Gorge and then converged with the upper Changjiang/Yangtze River at Shigu town, Yunnan province [10] (Figure 1). Due to multiple uplifts of the Tibetan and Yungui Plateaus (Appendix A), the south-flowing course of the palaeo-upper Changjiang/Yangtze River was blocked, and the Shuiluo River was captured by the middle Changjiang/Yangtze River at the second bend (SB) in the late early-Pleistocene [11]. Then, the flow of the upper Changjiang/Yangtze River along the Hutiao Gorge reversed from a southward to a northeastward direction [12]. However, the south-flowing courses of the upper Changjiang/Yangtze and Shuiluo Rivers have remained controversial and uncertain [9,13,14,15,16,17]. These south-flowing rivers, including the upper Yangtze, Shuiluo, and Yalong Rivers, are also thought to have emptied into different basins [16,17,18] (Appendix A). Furthermore, besides a course through the Red River to the South China Sea, the upper Yangtze River also likely flowed into the South China Sea through the Longchuan River or a tributary of the Mekong River [17,19]. Limited by different sampling localities, analytical methods, and timescales, contradictory results or conclusions of drainage evolution often occurred in different geological or geomorphologic studies.

However, aquatic species inhabiting the rivers or lakes provided a new opportunity to distinguish ambiguous drainage histories due to co-evolution between freshwater species and watershed [20,21,22,23,24,25,26,27]. Historical geologic events, such as tectonic uplifts and river rearrangements, can impact the spatial distribution and speciation of aquatic organisms [28]. Old continuous riverine habitats would be divided into isolated and fragmental ranges due to river capture. Genetic divergence and new lineages of aquatic species can subsequently occur due to accumulated gene mutations in these isolated riverine habitats after thousands of generations [29]. On the other hand, due to environmental difference or limited migration abilities, freshwater organisms from the same temporary river can still show community or genetic differences between tributary and newly connected main streams [27]. Conflicting aquatic species evolutionary relationships and present rivers connections provide us an opportunity to study ancient drainage reorganization.

Unlike traditional morphological characters, genetic data and the phylogenetic relationship reconstruction of closely related freshwater species can be used to elucidate the abovementioned drainage history and geographic associations between rivers [18,30]. Some evolutionary closely related lineages or species have often been identified from adjacent basins or drainages [18,22,23,31,32], which indicates that palaeo-connectivity and capture events are likely imprinted in the genetic information of the freshwater organisms. For some Chinese rivers originating from the Tibetan Plateau, the across-basins dispersal of freshwater fish has also been observed between the Qiadam basins and the Heihe River in Northwest China [24,33], between the middle Changjiang River and upper Yellow River [30,34], between the middle Changjiang River and Pearl River in South China [35], and between the lower Changjiang River and Han River in East China [35]. These biogeographic studies confirmed that palaeo-drainage rearrangements frequently co-evolved with freshwater fishes. Thus, genetic data show more advantage to recover drainage history relative to regional geologic or geomophologic data.

As freshwater fish endemic to the Tibetan Plateau, subfamily Schizothoracinae includes three groups corresponding well to three phases of uplifts of the Tibetan Plateau with different elevation distributions: primitive grade (PG; genera *Racoma*, *Schizothorax* and *Aspiorhynchus*; 500–3500 m), specialized grade (SG; genera *Ptychobarbus*, *Gymnodiptycus* and *Diptychus*; 2750–3750 m), and highly specialized grade (HSG; genera *Gymnocypris*, *Oxygymnocypris*, *Schizopygopsis*, *Platypharodon* and *Chuanchia*; 3750–4750 m) [36]. They are predominant species with slow growth and physiological adaptations to cold and hypoxic water of the upper and middle reaches of drainages in the Tibetan Plateau [31]. These fishes are ideal species to investigate the co-evolution of drainage dynamics and the freshwater biota.

In this study, a member of HSG *Schizopygopsis* fish from the upper Changjiang/Yangtze River (i.e., Jinsha River) and its tributaries (e.g., Shuiluo, Litang, and Yalong Rivers) was sampled to reveal co-evolution between fish and rivers due to its dependence on a riverine habitat with the highest altitude. Because of the misleading taxonomy of schizothoracine fish in earlier studies [23,32], this study will focus on the biogeographic implications of different genetic clades from the upper Yangtze River instead of species identification and the interspecific phylogenetic relationship of *Schizopygopsis* fish. Given the aforementioned uncertainties on the palaeo-upper Changjiang/Yangtze River’s flowing courses, this integrative approach of utilizing a comparison of geological and biological clues may prove useful in resolving the ambiguous evolutionary history of the upper Changjiang/Yangtze River in this area.

## 2. Materials and Methods

### 2.1. Sampling and Data Acquisition 

A total of 53 freshwater fishes (*Schizopygopsis microcephalus*) were collected using a casting net with a mesh size of 5 mm from the first bend (FB) of the upper Changjiang/Yangtze/Jinsha River at Shigu town, Yunnan province, in 2017 (Table 1, Figure 2). From each individual, ten grams of muscle tissue were preserved in 95% ethanol. The capture and sampling of individuals were undertaken in accordance with an animal research ethics permit (W20211202) granted by the East China Normal University Animal Ethics Committee. Total DNA was extracted from each sample using a standard phenol–chloroform approach [37]. The full-length segment of mitochondrial (mtDNA) cytochrome *b* (*Cyt b*) gene was amplified via PCR using the primers L14724 and H15915 [38]. Amplifications were conducted in 25 uL volumes containing 5 uL of template DNA, 1 × PCR reaction buffer, 2 mM MgCl_2_, 300 nM each of primers L14724 and H15915, 200 mM dNTPs, 0.5 unit of Taq polymerase, and ddH_2_O. Reactions were conducted under the following PCR conditions: an initial denaturation at 94 ºC for 4 min, followed by 40 cycles of 94 °C for 35 s, 52 °C for 40 s, 72 °C for 70 s, and a final extension of 72 °C for 8 min. Successfully amplified products were separated on a 1.5% agarose gel and purified with the Gel Extraction Mini Kit (Watson BioTechnologies, Shanghai, China). Purified products were sequenced in both directions using the primer pairs L14724 and H15915 on an ABI Prism 3730 automatic sequencer. Near complete sequences (1138 bp) of cytochrome *b* were aligned in Cluster-X [39] using default parameters. 

A total of five unique *Cyt b* haplotypes were identified from the collected samples of *S. microcephalus* in the first bend (FB) of the upper Changjiang/Yangtze River and deposited into GenBank under accession numbers MN399193-MN399197. To assess the evolutionary relationship of the five *S. microcephalus* haplotypes with the same species or other species of *Schizopygopsis* in the Tibetan Plateau [32], the top 12 most similar *Cyt b* sequences (percent identity > 97.98%) were supplemented through BLAST searches in GenBank (Table 1, Figure 2): viz. 1 haplotype of *S. microcephalus* (KY461332) from the Tuotuo River (upper Changjiang/Yangtze/Jinsha River); 2 haplotypes of *S. malacanthus malacanthus* (DQ533793, DQ533794) from the Litang River; a tributary of the Yalong River (Yu et al., 2006); 4 haplotypes of *S. malacanthus malacanthus* (DQ533789-DQ533792) from the Shuiluo River, which converges to the Changjiang/Yangtze/Jinsha River at the second bend (SB); 3 haplotypes of *S. malacanthus* (DQ309360, DQ646898, DQ646899) from the upper Yalong River in Chengduo county, Qinghai province; 1 haplotype of *S. thermalis* (DQ309367) from the upper Changjiang/Yangtze/Jinsha River (Tuotuo River), and 1 haplotype of *C. labiosa* (KT833098) from the upper Yellow River.

Furthermore, 33 additional *Cyt b* sequences of schizothoracine fish retrieved from GenBank were also integrated (Appendix A): e.g., 3 haplotypes (T603, T604, T605) of *C. labiosa* (KY461371-KY461373), 4 haplotypes (Gech31, Q1, Q2, Q3) of *G. eckloni chilianensis* (KM371149, KY461311-KY461313), 6 haplotypes (T63, T64, T65, T250, T251, T252) of *G. namensis* (KY461381-KY461383, KY461335-KY461337), 3 haplotypes (T521, T529, T548) of *S. pylzovi* (KY461363, KY461364, KY461367), 3 haplotypes (SkT260, T261, T262) of *S. kialingensis* (KY461338-KY461340), 3 haplotypes (T290, T291, T292) of *S. malacanthus* (KY461344-KY461346), 2 haplotypes (T240, T242) of *S. microcephalus* (KY461331, KY461333), 3 haplotypes (T270, T271, T272) of *S. anteroventris* (KY461341-KY461343), 3 haplotypes (T150, T151, T152) of *S. thermalis* (KY461318-KY461320), and 3 haplotypes (T623, T624, T625) of *S. malacanthus chengi* (KY461378-KY461380). These sequences covered mitochondrial clades B1a, B1b, B3, B4, and C2b found by Tang et al. [32]. The three haplotypes of *S. malacanthus chengi* (T623, T624, T625) act as an outgroup in the subsequent phylogenetic reconstruction of dataset I that includes 50 haplotypes. Then, phylogenetic trees of dataset II composed of the abovementioned 17 top similar haplotypes with two outgroups (e.g., SkT260, Gech31) were also further inferred.

### 2.2. Phylogenetic Reconstruction

The appropriate evolutionary models of two datasets were chosen by AIC in Modeltest 3.6 [46]. The model parameters of the two datasets are listed as follows: (1) Dataset I (GTR + G), base frequencies with A (0.2641), C (0.2665), G (0.1624); rate matrix with R [A-C] =1.6899, R [A-G] = 125.6114, R [A-T] = 0.8044, R [C-G] = 7.7435, R [C-T] = 33.6950, R [G-T] = 1.0000, γ distribution shape parameter G= 0.1158. (2) Dataset II (GTR + I), base frequencies with A (0.2559), C (0.2670), G (0.1707); rate matrix with R [A-C] = 0.6143, R [A-G] = 45.7215, R [A-T] =0, R [C-G] = 6.1788, R [C-T] = 23.2917, R [G-T] =1.0000; proportion of invariable sites (I) = 0.8103.

Phylogenetic relationships were inferred among the *Cyt b* haplotypes of two datasets via Maximum Parsimony (MP) [47], Maximum Likelihood (ML) [48], and Bayesian inference (BI) [49] methodologies in PAUP 4.0 [50], PHYML 3.0 [51], and MrBayes 3.2 [52], respectively. For the MP and ML trees of the two datasets, nodal supports were assessed using 10000 bootstrapping replicates [53]. BI trees of two datasets were also performed using the following settings: (1) Dataset I, ngen = 10,000,000, samplefreq = 1000, burnin = 2500, Nchains = 4, and Nruns = 2. (2) Dataset II, ngen= 100,000,000, samplefreq =10,000, burnin = 2500, Nchains = 4, and Nruns = 2. The convergences of independent runs were achieved when white noise was seen in the overlay plot of the generation versus the log-likelihood probability for both runs with a lower average standard deviation of split frequencies (dataset I, 0.003213; dataset II, 0.003314). The nodal stability in the BI trees was assessed by posterior probability (PP). Due to weak divergence among different haplotypes or species, evolutionary relationships of ingroups in dataset II were also assessed using an intraspecific network [54] in NETWORK (version 5.0.1.1; fluxusengineering, 2018).

### 2.3. Estimation of Divergence Time and Biogeographic Analyses

The divergence time between each pair of clusters or subclusters from 17 closely related sequences in this study (Clade D) was estimated in BEAST v.2.6.7 [55] based on dataset II. The GTR + I evolution model was used with empirically determined base frequencies. For the tree prior parameter, the Speciation: Birth–Death process [56] was used and modeled with an uncorrelated lognormal relaxed clock [57]. As no fossil record of the schizothoracine fish was available, this study utilized a well-dated geological event- the “Kunlun-Huanghe movement”, which occurred at ca. 1.1–0.7 Ma and led to the separation of the upper Yellow River from the upper Yangtze River [33,58] to calibrate the divergence. A normal distribution prior was applied to estimate the most recent common ancestor of Clade DI from upper Yellow, Tuotuo, and Yalong Rivers to a mean age of 0.9 Ma. Three independent MCMC runs of 200 million generations with a sampling frequency of every 20,000 generations were conducted. The parameter values from runs with ESS over 200 were combined to summarize posterior divergence times and 95% highest posterior density limits in Tracer v1.6 [59] after discarding 10% of the first trees as burn-in.

The Bayesian binary method (BBM) implemented in RASA 4 [60] was used to reconstruct the ancestral area in data set II (Clade D). The geographic data were based on the present distribution of several *Schizopygopsis* species (Figure 2 and Table 1). The BBM analysis was run for 100 million cycles with 10 chains and a sampling frequency of every 10,000 cycles through discarding 20% samples. The temperature was set at 0.1, and a fixed JC model was used. The maximum number of areas for each node was four. The information of each node was summarized and plotted as a pie chart.

## 3. Results

### 3.1. Phylogenetic Analyses and Divergence Time Estimation

Through integrating samples from this study and Tang et al. [32] (Dataset I), consistent phylogenetic topographies were revealed by three tree building methodologies (MP, ML, and BI) (Appendix A). Except clade C1 of *Schizopygopsis*, most other mitochondrial clades (e.g., B1a, B1b, B3, B4, and C2b) occurring in Tang et al. [32] were recovered in the present phylogenetic reconstruction. Unlike the observed monophyletic clade (B3) of *Schizopygopsis* in Tang et al. [32], paraphyly of B3 was revealed in this study. A new clade D nested in B3 was composed of three well-resolved genetic clusters (DI, DII, and DIII) with strong nodal supports (Figure 3 and Appendix A). Clade DI included samples of *S. malacanthus* from the upper Yalong River, *S. thermalis* from headwater of the Tuotuo River (upper Changjiang/Yangtze/Jinsha River), and *C. labiosa* from the upper Yellow River. Clade DII was composed of samples of *S. malacanthus malacanthus* from the Shuiluo River, a tributary converging with the Changjiang River (Yangtze/Jinsha River) at the SB (Figure 2). Clade DIII comprised two subclusters (Clade DIIIa and DIIIb). Among them, Clade DIIIa included five haplotypes (YNSG08, 23, 27, 31, 65) from the FB of Changjiang/Yangtze River at Shigu Town, Yunnan province, and a haplotype (HmT241) of *S. microcephalus* from the Tuotuo River (upper Changjiang/Yangtze/Jinsha River). And Clade DIIIb consisted of two haplotypes (Smm5, 6) from the Litang River, a tributary of the Yalong River (Figure 2). Although the ancestral node of Clade DI and clade DII obtained a weak nodal support (59/69/0.84) in Dataset I (Appendix A), the grouping of Clade DI and Clade DII was not observed in Dataset II (Figure 3). The evolutionary network of 17 closely related haplotypes from clade D (Dataset II) recovered the same genetic clusters and subclusters (DI, DII, DIIIa and DIIIb) as those observed in phylogenetic trees (Figure 3, Figure 4, and Appendix A). Haplotypes from the upper Yalong (SmZ1 and SmFEM2) and Litang Rivers (Smm5 and Smm6) are basal or interior relative to other haplotypes in Clades DI and DIII, respectively (Figure 3 and Figure 4). 

No obvious biogeographic clusters corresponding to the contemporary watersheds of the Yellow, Changjiang/Yangtze/Jinsha, and Yalong Rivers (Figure 2) were observed, and the haplotypes from the FB of upper Changjiang/Yangtze River at Shigu Town (Clade DIIIa) were more closely related to those from the Litang River (Clade DIIIb) than to those from the closer tributary, the Shuiluo River (Clade DII). Furthermore, samples from the Litang River (Clade DIIIb) were not closely related to those from the adjacent main stream, the upper Yalong River (Clade DI). 

The coalescent time of clade D was inferred to be 1.56 Ma and coalescent times for Clade DI and DIII occurred at 0.87 and 0.90 Ma, respectively. The coalescent times of Clade DII, Clade DIIIa, and Clade DIIIb were similar to each other, which ranged from 0.46 to 0.50 Ma. Furthermore, samples from the Yalong River coalesced with those from the upper Yellow River and Yangtze/Tuotuo River at a similar 0.34 Ma, respectively (Figure 3).

### 3.2. Ancestral State Reconstruction

BBM revealed six events of dispersal and six events of vicariance along the phylogeny of clade D (Dataset II) (Figure 5). The ancestral region of three clusters in clade D (node 35) was inferred to be Yalong River (PP = 59.86). Additionally, nodes 31 and 33 indicated that the Yalong River acted as an origin of dispersal to the upper Yellow River (PP = 96.59) and Yangtze/Jinsha/Tuotuo River (PP = 90.91), respectively. Node 30 indicates that the common ancestor of Clades DII and DIII likely originated from the Shuiluo River (PP = 63.53). Node 26 indicates that Clade DIII has a higher probability to have originated from the Litang River (PP = 40.68) than from Shigu town, Jinsha River (PP = 36.12). The ancestral population of Clade DIIIa (node 24) was inferred to originate from Shugu town, Jinsha River (PP = 94.59).

## 4. Discussion

### 4.1. Taxonomic Assignments

A haplotype of *C. labiosa* (Clabi1) from the upper Yellow River was found to be nested in Clade DI with haplotypes of *S. malacanthus* and *S. thermalis* from the upper Yalong and Tuotuo Rivers/Changjiang/Yangtze/Jinsha River (Figure 3, Figure 4, and Appendix A). Closely related fish species are often found in neighboring drainage systems [18,22,32] and in this case could be caused by a number of factors: (1) incomplete lineage sorting or ancestor polymorphism due to insufficient separation time; (2) hybridization between *C. labiosa* and *Schizopygopsis* spp.; (3) misidentification of species due to morphological variation of a single species with a wide distribution [61]. Due to the long physical isolation (ca. 1.1–0.7 Myr) between the Changjiang/Yangtze and Yellow Rivers since the Kunlun-Huanghe (Kunhuang) Movement [33,58] and the deep divergences observed among different clusters of *S. malacanthus* (Figure 3, Figure 4, and Appendix A) or between Clabi1 and other haplotypes of *C. labiosa* in B3a from the upper Yellow River (Appendix A), the latter two explanations are more plausible than ancestor polymorphism. However, hybridization can also be eliminated because *C. labiosa* showed a consistent, close relationship with *S. pylzovi* in clade B3 of *Schizopygopsis* based on genome-wide SNP and mitochondrial gene study [32]. Due to convergent evolution in Tibetan Plateau rivers with similar selection pressures, morphological traits frequently mislead real phylogenetic relationships and fish taxonomy. *Chuanchia labiosa* has been treated as *S. labiosa* [32]. Similar phenomena were also observed in *S. microcephalus*, which had been misidentified as *Herzensteinia microcephalus* [24,61,62]. Furthermore, except Oxygymnocypris, two other genera (*Gymnocypris*, *Platypharodon*) of HSG were also considered as synonym to genera *Schizopygopsis* [32]. Thus, this study aimed to reveal the biogeographic implications of clade D (Dataset II of *Schizopygopsis* fish) rather than species taxonomy and interspecific evolutionary relationship.

### 4.2. Palaeo-Drainages Connection Signature

Due to limited dispersal abilities across land, phylogenetic relationships of freshwater fish can be used to reconstruct the evolutionary history of river drainages [21,26]. Based on the phylogenetic relationships and ancestral origin inference of *Schizopygopsis*’ clade D haplotypes (Figure 3, Figure 4, and Appendix A), novel clues on historical drainage connectivity events in the upper Changjiang/Yangtze and Yellow systems were indicated. 

Ancient, rather than contemporary, drainage connections of upper Changjiang/Yangtze and Yellow systems were implicated from Clades DI and DIII (Figure 6). The closely related haplotypes and shared lineages across adjacent rivers in Clade DI likely indicate historical gene flow across the connected palaeo-drainages among the Tuotuo (upper Changjiang/Yangtze/Jinsha River), Yalong, and Yellow Rivers (Figure 3, Figure 4, Appendix A and Figure 6a). Furthermore, the dispersal directions were inferred to originate from the upper Yalong to the adjacent Changjiang/Yangtze/Jinsha/Tuotuo and Yellow Rivers (Figure 5 and Figure 6a). Trans-watershed genetic links were also observed in freshwater fishes of schizothoracine and *Triplophysa* between contemporary isolated Chinese northern drainages (e.g., Shule River, Heihe River, Yellow River, and Chaidam Basin) and southern tributaries of the middle Changjiang/Yangtze River (e.g., Dadu River, Min River, Jialing River, etc.) [24,30,33,63]. Similar types of ancient river connectivity have also been revealed by phylogenetic trees and/or population genetic structures of freshwater species from USA [64,65], Brazil [66,67], New Zealand [27,68], and Australia [25]. Due to tectonic activity and geomorphological changes caused by the Tibetan Plateau uplifts, a single river or basin could have been separated into different sections [2]. Consequently, freshwater species inhabiting previous old rivers or continuous watersheds would be isolated into fragmented habitats, and population divergence or lineage differentiation would have subsequently occurred due to synchronously accumulated DNA substitutions between these physically separated habitats [29,69]. 

Western China was flooded by the Tethys Sea before 40 Ma [70,71]. The India–Asia collision caused the upheaval of the Asian plate and sea regression in southwestern China 32 Ma [72]. The Tibetan Plateau experienced two cycles of uplift and plantation with an altitude between 1000 and 2000 m during 40–3.6 Ma [73] (Appendix A). Three periods of abrupt upheavals subsequently occurred (Appendix A): the Qinghai-Xizang Movement with an average height of 2000 m during 3.6–1.7 Ma, the Kunhuang Movement with an average height of 3000 m during 1.1–0.7 Ma, and the Gonghe Movement reaching the present altitude of 4000–5000 m ca. 0.15–0 Ma [73]. A number of west-flowing or south-flowing palaeo-rivers (e.g., Chaidam basin, Shule River, Heihe River, upper Yellow River, upper Yalong River, upper Changjiang/Yangtze/Jinsha River, Dadu River, Min River, and Jialing River) likely connected with each other before these times of uplifts. The Kunhuang Movement caused the headwater capture of the Yellow River in the northeast Tibetan Plateau [74]. The Gonghe Movement caused the separation of the Qinghai Lake from the Yellow River, and the middle Yellow River was captured by the lower Yellow River cutting through the Longyang and Sanmen Gorges. By using a calibrated point of the Yangtze River–Yellow River’s last connection at ca. 0.9 Ma, the last migration time from the Yalong to upper Changjiang/Tuotuo or Yellow Rivers consistently occurred at 0.34 Ma (Figure 3). The interglacial warmer climate and strong summer monsoon during 0.32–0.34 Ma would have caused heavy precipitation and promoted *Schizopygopsis* fish’s dispersal from the Yalong to neighboring Tuotuo and Yellow Rivers [75,76].

Palaeo-drainage connections were also indicated in Clade DIII (Figure 3, Figure 4 and Figure 5). Although the contemporary Litang River only indirectly drains into the Changjiang/Yangtze/Jinsha River through the lower Yalong River (Figure 2), the Clade DIIIb from the Litang River exhibited a sister relationship with Clade DIIIa from the upper Changjiang/Yangtze River (Tuotuo-Jinsha River). Furthermore, the Litang River probably acted as an evolutionary origin of Clade DIII (Figure 5). The historical migration route of fish in Clade DIII could not be from the Litang River through the lower Yalong River to Changjiang/Yangtze/Jinsha River because substantial genetic differentiation (D_net_ = 4.22 × 10^−3^) between the Litang and Changjiang/Yangtze/Jinsha Rivers (DIIIb and DIIIa) would more likely have occurred in isolated habitats. The most probable historical dispersal pathway is from the headwaters of the Litang River to the upper Changjiang/Yangtze/Jinsha River through the Dingqu or Shuoqu Rivers, two tributaries of the Changjiang/Yangtze/Jinsha River across a short distance (Figure 6b). Given the adaptation of *Schizopygopsis* to low temperatures in high-altitude (4000–5000 m) rivers [31] (Table 1), it is highly possible to have had dispersal between connected headwaters of adjacent rivers [22]. 

Genetic connectivity across drainages can be influenced by natural events (e.g., hurricane) or by animal mediated transportation (e.g., birds or anthropogenic carriage). However, it is highly improbably that hurricanes and/or birds had a significant impact on the dispersal and distribution of Schizopygopsis fish. Sucessfully colonizing a new ravine habitat would be very challenging with only a small number of fish. Furthermore, genetic diversity of such small populations would be quickly eroded due to genetic drift [77]. Additionally, overfishing and habitat loss have greatly reduced native schizothoracine fish stocks in the Tibetan Plateau since 1950 [31]. Artificial breeding and release of native fishes are expected to play a positive role in natural population restoration. To date, only three species of *Schizopygopsis* fish including *S. pylzovi*, *S. younghusbandi*, and *S. microcephalus* have been successfully bred artificially and hatched during the past five years [78,79,80]. The artificial propagation of *S. microcephalus* was achieved based on parent fish caught from the southern headwaters (Dangqu River) of Yangzte River at 4819 m above sea level [79]. Generally speaking, newly released or founded populations will show lower genetic diversity or younger haplotypes relative to the source population due to the founder effect [29,81,82]. If the hatched fries of *S. microcephalus* had been transported to other localities in the Yangtze River, an ancestral origin from the Tuotuo River would have been observed. However, contrary to the hypothesized artificial release, the only haplotypes (HmT241) of *S. microcephalus* from the Tuotuo/upper Yanggtze/Changjiang/Jinsha River in this study were nested in those from the FB at Shigu town (Figure 2, Figure 3 and Figure 4). Furthermore, the dispersal direction of *S. microcephalus* within Clade DIIIa was also inferred to migrate from the FB to Tuotuo River (Figure 5). Thus, the potential release of these artificially cultured fries did not influence the biogeographic history inference of *Schizopygopsis* fish with earlier samples of *S. microcephalus* and other non-cultured species in this study. 

As mentioned above, the upper Yalong, Shuiluo, and Litang Rivers were indicated to act as an origin of dispersal or endemic habitat in Clades DI, DII, and DIII (Figure 5). These genetic clusters likely originated from an ancestral population and were subsequently differentiated from each other due to physical vicariance and habitat fragmentation [29]. A giant ancient lake or basin across these adjacent drainages including the Shuiluo, Litang, and Yalong Rivers can be inferred from the evolutionary relationship and spatial distribution of these fishes (Figure 6b). Geologic clues suggest that a giant Bashu Lake and a smaller Xicang Lake connecting the west-flowing palaeo-middle Changjiang/Yangtze River and other local rivers may have occurred in this area during the Jurassic and Cretaceous [83]. The palaeo-lakes likely persisted during the Cretaceous and Tertiary because of the low elevation of the Tibetan Plateau (less than 1000 m) (Appendix A). The ancestral population of *Schizopygopsis’* clade D probably inhabited one of these palaeo-basins before 1.56 Ma (Figure 3 and Figure 6). Subsequently, these palaeo-lakes may have become gradually smaller and been divided into different drainages during the abrupt uplifts of the Tibetan Plateau during the late Tertiary and Quaternary [6] (Appendix A). Present-day molecular data shed light on the evolutionary influence of the Qinghai-Xizang and Kunhuang Movements on drainage changes and lineage diversification in the eastern marginal of the Tibetan Plateau (Figure 3 and Appendix A).

### 4.3. River Capture History in the Upper Changjiang System

The close relationship of Clade DIIIa and Clade DIIIb between the Litang and upper Changjiang/Yangtze/Jinsha Rivers further suggests a recent capture/reversal of the Litang River to its modern south-flowing course into the lower Yalong River in 0.8963 Ma (Figure 2 and Figure 3). Although the contemporary Litang River drains into the Yalong River (Figure 2), the fish from the Litang River exhibit deep genetic divergence with those of the upper Yalong River (Figure 3 and Figure 4), which indicates a long period of physical separation between the Litang River and Yalong River (Figure 6b). The separation from the upper Yangtze River and subsequent capture of the Litang River by the lower Yalong River was likely caused by the recent Kunhuang Movement and Tibetan Plateau uplift during 1.1–0.7 Ma (Figure 3 and Appendix A).

The haplotypes (Smm1, 2, 3, and 4) of *S. malacanthus malacanthus* from the Shuiluo River (Clade DII) showed deep divergence with the upper Changjiang/Yangzte/Jinsha River subcluster (Clade DIIIa) from the FB at Shigu town and Tuotuo River (Figure 2, Figure 3 and Figure 4), which also suggests an ancient separation of the Shuiluo River from the upper Yangtze River and a recent river capture at the second bend (SB) of the Changjiang (Figure 6b). Some geologists believe that the palaeo-Shuiluo River flowed south through the Hutiao Gorge and converged with the upper Changjiang/Yangtze/Jinsha River at Shigu town [11] (Figure 1). If this hypothesis had been true, the Shuiluo River and upper Changjiang/Yangtze/Jinsha River would have shared the same genetic clade or cluster because there would have been no physical barrier to prevent gene flow between them. However, the present DNA molecular data do not support this convergence hypothesis of Shuiluo River and the upper Changjiang/Yangtze/Jinsha River in the FB (Figure 3 and Figure 4). Although some authors have insisted that the upper Changjiang/Yangtze/Jinsha River flowed south through Shigu, Jianchuan Basin, to the Red River [9], there are no found fluvial sedimental records along this route [15]. Recently found fluvial sedimental evidence in Tongdian, Madeng, and Nanjian basins indicates that the upper Changjiang/Yangtze/Jinsha River likely flowed southward into the Red River or Mekong River along a western course instead of along Shigu town and Jianchuan basin [84]. Furthermore, the upper Changjiang/Yangtze/Jinsha River, Shuiluo River, and Yalong River were even thought to be independent palaeo-rivers flowing south into three basins along different courses (Appendix A), and the palaeo-Shuiluo River was believed to flow south through the right section instead of the left section (Hutiao Gorge) around the SB (Sanjiangkou) [16]. The biogeographic distribution of Tibetan Plateau fish in the Yangtze River also provides another clue to reveal Shuiluo River’s ancient flowing course. A biogeographic boundary was found in the Hutiao Gorge [61]. The proportion of Tibetan Plateau fish abruptly declined from the upper reach (54%) to the lower reach (18%) of the Yangtze River around the Hutiao Gorge. Thus, integrated evidence, including molecular, biogeographic, and geologic data, consistently suggests that the Hutiao Gorge likely played a role as a dispersal barrier of aquatic species instead of a bridge between the upper Yangtze River and Shuiluo River/lower Yangtze River (Figure 6b). Future work will still be needed to test the three parallelly south-flowing rivers model by integrating more biological and geologic evidence (Appendix A).

## 5. Conclusions

This study provides DNA molecular evidence for the ancient connectivity of contemporary isolated drainages (e.g., the upper Yalong, Changjiang/Yangtze, and Yellow Rivers, or the Litang and upper Changjiang/Yangtze/Jinsha Rivers). One or more giant palaeo-lakes across contemporary Shuiluo, Litang, and Yalong Rivers are suggested to have existed and acted as the ancestral origin of *Schizopygopsis* fish (Clade D) in the eastern margin of the Tibetan Plateau. Furthermore, the capture of the Litang River by the Yalong River and the capture of the upper Changjiang/Yangtze River by the Shuiluo River are also indicated due to the deep genetic divergence observed between these tributaries and mainstreams. As a preliminary analysis, this study demonstrates the utility of maternal phylogeographic studies of freshwater species in disentangling uncertain geologic, historical events and reconstructing the palaeo-flowing courses of contemporary rivers. A future comparative phylogeographic study of different freshwater species from the Tibetan Plateau based on more samples and nuclear genome will provide much more valuable information with high resolution to reconstruct Asian rivers’ evolutionary history.

## Figures and Tables

**Figure 1 genes-15-01148-f001:**
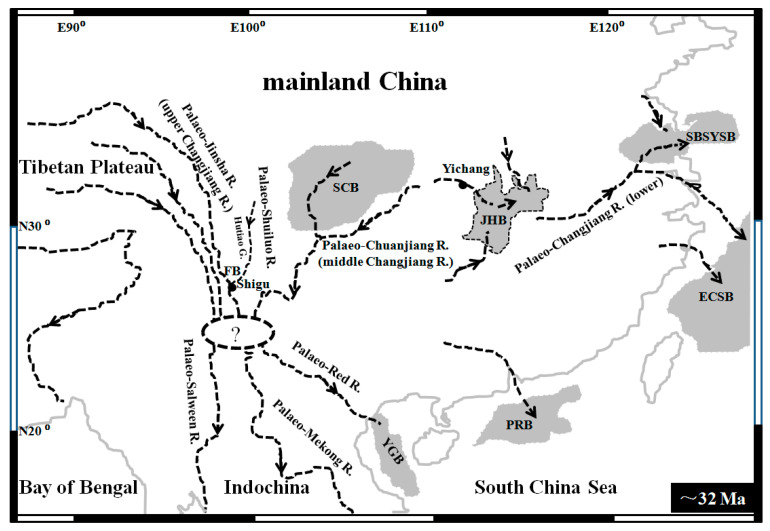
Map of palaeo-Changjiang drainage system 32 Ma. The palaeo-Changjiang River was divided into three unconnected main segments including the palaeo-Jinsha (upper Changjiang/Yangtze River), palaeo-Chuanjiang (middle Changjiang/Yangtze River), and lower Changjiang/Yangtze River. Several local rivers flowed into the Jianghan Basin. The flowing courses and directions of these palaeo-rivers are indicated by dashed lines with arrows. Ancient basins are shown in dark grey shadow. The contemporary shorelines of the mainland and islands are indicated by gray lines. SCB—Sichuan Basin, JHB—Jianghan Basin, SBSYSB—Subei-South Yellow Sea Basin, ECSB—East China Sea Basin, PRB—Pearl River Estuary Basin, YGB—Yinggehai Basin, FB—First Bend, R.—River, G.—Gorge. The broken oval region indicates ambiguous connections and flowing courses between the palaeo-Jinsha River, palaeo-Chuanjiang River, palaeo-Red River, palaeo-Mekong River, and palaeo-Salween River.

**Figure 2 genes-15-01148-f002:**
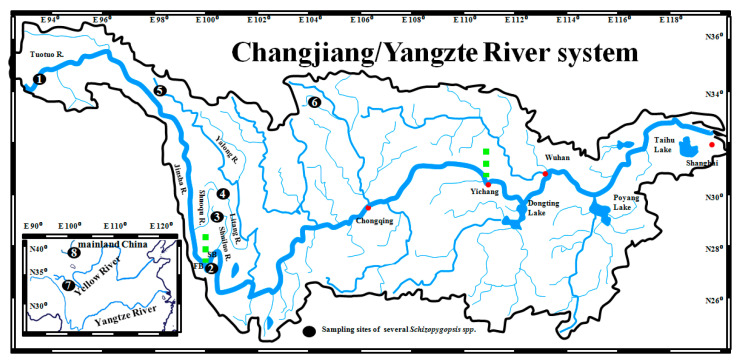
Sampling locations of *Schizopygopsis* fish from the contemporary Yellow River and upper Changjiang/Yangtze River systems. The three main segments of the Changjiang River (i.e., upper, middle, and lower reaches) are delineated by green dotted lines. Some main cities along the Changjiang/Yangtze River are indicated by red dots. R.—River, FB—First Bend, SB—Second Bend. The number of sampling localities is same as that shown in Table 1.

**Figure 3 genes-15-01148-f003:**
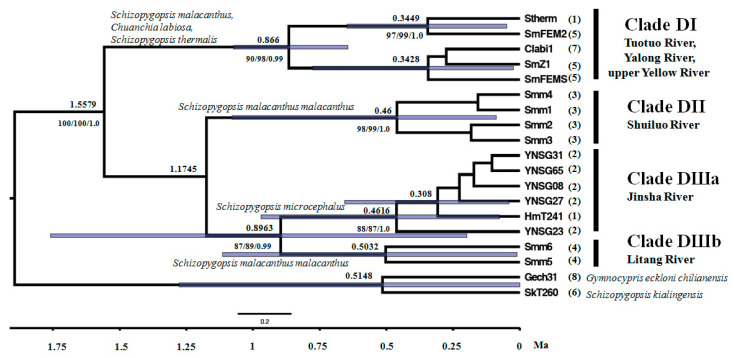
Bayesian relaxed clock cladogram based on cytochrome b of *Schizopygopsis*’ clade D. Blue bars indicate the highest posterior density of nodal ages. Numbers above branches show the median age of ancestral nodes, and numbers below branches indicate nodal support values measured as bootstrap values (BP) and posterior probability (PP) inferred by Maximum Parsimony (MP), Maximum likelihood (ML), and Bayesian inference (BI) methodologies, respectively. Numbers in brackets indicate sampling localities, which are also shown in Table 1 and Figure 2.

**Figure 4 genes-15-01148-f004:**
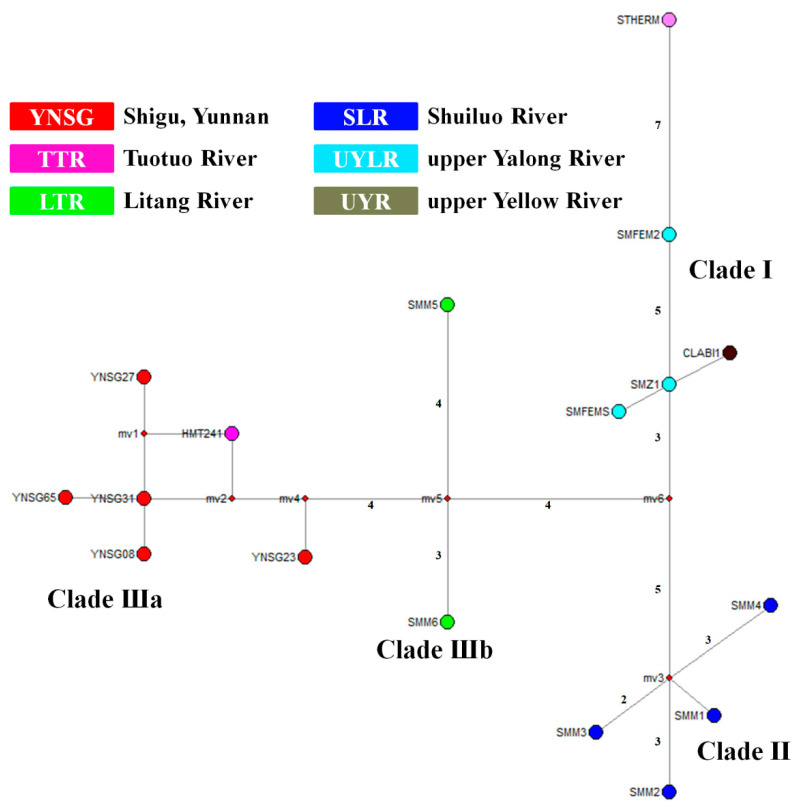
Median-joining network depicting relationships among cytochrome b haplotypes of *Schizopygopsis*’ clade D. Values along lines indicate substitution steps between haplotypes. Three genetic clusters (DI, DII, DIII) and two subclusters (DIIIa, DIIIb) are revealed.

**Figure 5 genes-15-01148-f005:**
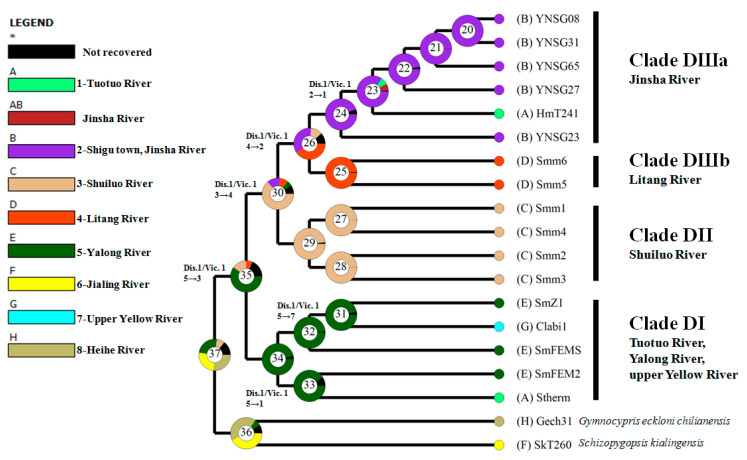
Ancestral area reconstruction of *Schizopygopsis*’ clade D. Pie charts at each node show posterior probabilities of an alternative ancestral distribution using different colors. Arrows show dispersal direction. The number of sampling localities from the legend is also shown in Figure 2. Vic, vicariance; DIS, dispersal. Star “*” indicates lumped ranges with a hidden probability of less than 5%.

**Figure 6 genes-15-01148-f006:**
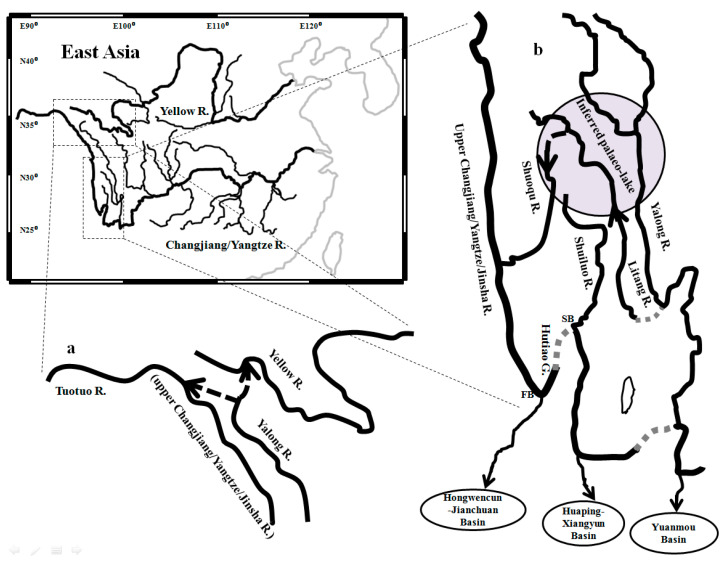
Palaeo-flowing course reconstruction of the upper Changjiang and Yellow Rivers’ system inferred from phylogenetic relationship and the ancestral origin of *Schizopygopsis*’ clade D in East Asia. (**a**) The ancient links among the upper Yalong River, Yellow River, and Changjiang/Yangtze/Jinsha/Tuotuo Rivers. (**b**) The palaeo-flowing courses and connectivity of the upper Changjiang/Yangtze/Jinsha River and Shuoqu, Shuiluo, Litang, and Yalong Rivers. The Litang River was suggested to have likely been connected directly to the Changjiang/Yangtze/Jinsha River instead of the Yalong River through the adjacent Shuoqu River. Three south-flowing rivers, upper Changjiang/Yangtze/Jinsha, Shuiluo, and Yalong Rivers, likely parallelly flowed into three different basins [16]. The broken arrows indicate ancient connectivity and the dispersal direction of fish among the upper Yellow, Changjiang, and Yalong Rivers or between the Litang and Shuoqu Rivers. The gray dotted lines in (**b**) indicate contemporary connections and ancient separations between the upper Changjiang/Jinsha River and Shuiluo River, Shuiluo River and Yalong River, and Litang River and Yalong River, respectively. The dark gray circle indicates an inferred palaeo-lake in the upper Shuiluo, Litang, and Yalong Rivers.

**Table 1 genes-15-01148-t001:** Sampling drainages, localities, and haplotypes information of *Schizopygopsis* fish from the eastern Tibetan Plateau.

Drainage (Locality)	Elevation (m)	Species	Haplotype	GenBank Number	Reference
**Changjiang/Yangtze System**					
1. Tuotuo River (Tanggula town, Qinghai, China)	4700	*Schizopygopsis thermalis* *S. microcephalus*	Stherm HmT241	DQ309367 KY461332	[40,41]
2. Jinsha River (First Bend, Shigu town, Yunnan, China)	1850	*S.* *microcephalus*	YNSG08, 23, 27, 31,65	MN399193-MN399197	this study
3. Shuiluo River (Litang and Daocheng Counties, Sichuan, China)	3976	*S. malacanthus malacanthus*	Smm1, 2, 3, 4	DQ533789-DQ533792	[42]
4. Litang River (Litang County, Sichuan, China)	3685	*S. malacanthus malacanthus*	Smm5, 6	DQ533793, DQ533794	[42]
5. Yalong River (Chengduo county, Qinghai, China)	3831	*S. malacanthus*	SmZ1, SmFEMS, SmFEM2	DQ309360, DQ646898, DQ646899	[40,43]
6. Jialing River (Têwo county, Gansu, China)	1700	*S. kialingensis*	SkT260	KY461338	[32]
**Yellow River System**					
7. Yellow River (Maduo county, Qinghai province)	4300	*Chuanchia labiosa*	Clabi1	KT833098	[44]
8. Heihe River (Zhangye city, Gansu, China)	3633	*Gymnocypris eckloni chilianensis*	Gech31	KM371149	[45]

## Data Availability

The data (DNA sequences) that support this study are available and have been deposited in GenBank with accession numbers: MN399193-MN399197.

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
