# Peer review of "Genetic Signature of River Capture Imprinted in Schizopygopsis Fish from the Eastern Tibetan Plateau"

_genes, 2024, doi:10.3390/genes15091148_

Round 1

Reviewer 1 Report

Comments and Suggestions for Authors

I found this manuscript of a good overall quality, I suggest the authors better clarify in the abstract, introduction and discussion, and explain in the material and methods section the choice to work in the experimental phase only with Schizopygopsis microcephalus, while all the others congeners species were elaborated in-silico. This should be more evident in the manuscript and the discussion section.

Please fix all the figures that seem attached to the file (also covering line numbers), and some of them report strange details like a presentation (e.g. fig 6, bottom-left corner).

Best regards

Author Response

Comments 1: I found this manuscript of a good overall quality, I suggest the authors better clarify in the abstract, introduction and discussion, and explain in the material and methods section the choice to work in the experimental phase only with Schizopygopsis microcephalus, while all the others congeners species were elaborated in-silico. This should be more evident in the manuscript and the discussion section.

Response: Some artificial breeding, release and ecological background of Schizopygopsis as highly specialized grade of the Schizothoracinae in the Tibetan Plateau was added into the introduction and discussion, which can explain why we selected Schizopygopsis instead of other fish. Due to misled taxonomy caused by convergent evolution, this study didn't aim to study interspecific phylogenetic relationship of Schizopygopsis microcephalus and other Schizopygopsis species. Thus, the main target of this study focus on biogeographic implication of different genetic clades/clusters of Schizopygopsis fish instead of particular species. 

Commetns 2: Please fix all the figures that seem attached to the file (also covering line numbers), and some of them report strange details like a presentation (e.g. fig 6, bottom-left corner).

Response: These figures will be adjusted well and fixed before its final publishing.

Reviewer 2 Report

Comments and Suggestions for Authors

Dear authors,

Congratulation for your work results.

In my opinion for the improving of the clarity of you work you should:

Highlight the limitation of the method/integrated methods used.

Highlight the limits of information, data and conclusions (i.e. relatively small number of samples and sampling points). This will let other authors to use your paper as a base for new studies even in new directions, considered now secondary but with potential in the future.

Bring some secondary potential explanations (cause-effects) for some situations in the field letting open other potential explanations too. This will not diminish the value of your work and conclusions, by contrary will stress the complexity of the subject.

Make a final gap analysis for what should come after this preliminary approach and results.

All the best

Reviewer

Author Response

Comments 1: Highlight the limitation of the method/integrated methods used.

Response: Thanks for your helpful suggestions. The limitation of the method used in geologic studies and integrated methods including molecular, biogeographic and geologic data were highlighted in abstract, introduction and discussion.

Comments 2:  Highlight the limits of information, data and conclusions (i.e. relatively small number of samples and sampling points). This will let other authors to use your paper as a base for new studies even in new directions, considered now secondary but with potential in the future.

Response: The limits of information, data and conclusions from previous geologic studies were presented in introduction, and compared to our genetic data in discussion.

Comments 3: Bring some secondary potential explanations (cause-effects) for some situations in the field letting open other potential explanations too. This will not diminish the value of your work and conclusions, by contrary will stress the complexity of the subject.

Response: Thanks for your good comments. Some secondary potential explanations were added into discussion: e.g., natural (e.g., hurricane) or animals' transportation (e.g., birds or anthropogenic carriage) could also likely influence genetic links across drainages.

Comments 4:  Make a final gap analysis for what should come after this preliminary approach and results. 

Response: Based on further analysis, we add following similar sentences in abstract, introduction and conclusion "this integrative approach utilizing a comparation of geological and biological clues in this study may prove useful information in resolving the ambiguous evoltionary history of some geologic events".

Reviewer 3 Report

Comments and Suggestions for Authors

The study analyzed the genetic structure of a freshwater fish species from China proposing inferences for demographic history. The analysis is only based on one mitochondrial gene, a fact that is a limitation of the study. Additionally the number of the samples is relatively small. Nevertheless, the analysis is comprehensive and well documented, whereas the lineages and haplogroups discovered are interesting. 

Also the scope is not clear. Is it just to reveal the relationships of Schizopygopsis fishes? If general evolutionary history is targeted, then the approach is not correct. Different fish have very differential biological aspects and general conclusions for all the basin inhabitants are practically speculative. Thus I suggest to rewrite the scope, the title, the discussion and the general way of writing of the manuscript targeting only to the particular species or genus.

Specific comments

In the abstract, instead of referring to “several” Schizopygopsis fishes, better provide the precise number

A new lineage of which species? This has to be mentioned in the abstract

In general I disagree with the term “several”, I recommend to avoid it

In the MJ network of Figure 4, the circles should not be proportional of the times each haplotype was detected? Please chaeck and re-anayse if necessary

Finally, some socioeconomic data for the fish are missing. Is there fisheries developed for the analyzed fish? Is it important for any other reason. Please combine these data in the Introduction and Discussion

Author Response

Comments 1: The study analyzed the genetic structure of a freshwater fish species from China proposing inferences for demographic history. The analysis is only based on one mitochondrial gene, a fact that is a limitation of the study. Additionally the number of the samples is relatively small. Nevertheless, the analysis is comprehensive and well documented, whereas the lineages and haplogroups discovered are interesting. 

Response: Indeed, only one mitochondrial gene shows insufficient genetic information of population. As a preliminary study, this manuscript just try to study maternal history of Schizopygopsis fish through combining limited samples from ours and other authors. More genes and data will be added in the further study.

Comments 2:  Also the scope is not clear. Is it just to reveal the relationships of Schizopygopsis fishes? If general evolutionary history is targeted, then the approach is not correct. Different fish have very differential biological aspects and general conclusions for all the basin inhabitants are practically speculative. Thus I suggest to rewrite the scope, the title, the discussion and the general way of writing of the manuscript targeting only to the particular species or genus.

Response: Thanks for your help suggestions. The title was changed to "Genetic signature of river capture imprinted in Schizopygopsis fish from the eastern Tibetan Plateau". The scope and aim of this study focus on biogeographic implication of Schizopygopsis fish's genetic clusters instead of particular species/genus taxonomy, or interspecific phylogenetic relationship due to misled taxonomy caused by convergent evolution (Tang et al., 2019). The introduction and discussion were also revised to clearly clarify our study scope.

Comments 3: In the abstract, instead of referring to “several” Schizopygopsis fishes, better provide the precise number

Response: “several” was deleted from title, abstract and text.

Comments 4: A new lineage of which species? This has to be mentioned in the abstract

Response: " lineage" was replaced by "clades" of Schizopygopsis instead of particular species' lineage.

Comments 5: In general I disagree with the term “several”, I recommend to avoid it

Response: I have accepted your suggestion and delete the term “several” in this manuscript.

Comments 6:  In the MJ network of Figure 4, the circles should not be proportional of the times each haplotype was detected? Please chaeck and re-anayse if necessary

Response: This MJ network of Fig. 4 mainly aim to present haplotypes' evolutionary relationship of Schizopygopsis' clade D from different rivers. Some haplotypes retrieved from previous studies without frequency data, but which didn't influence our evolutionary relationship analysis in this study.

Comments 7: Finally, some socioeconomic data for the fish are missing. Is there fisheries developed for the analyzed fish? Is it important for any other reason. Please combine these data in the Introduction and Discussion.

Response: Evolutionary influence of natural (e.g., hurricane) or animals' transportation (e.g., birds or anthropogenic carriage) on genetic links amoung differen rivers was discussed, and artificial breeding and release of Schizopygopsis fish into the field were compared to our data in order to infer the reason of genetic connectivities between different rivers in discussion.

Round 2

Reviewer 2 Report

Comments and Suggestions for Authors

Congratulations for the work results.

More in deep analysis can be done on the subject in the future.

All the best.

Reviewer 3 Report

Comments and Suggestions for Authors

The authors addessed my comments, the study may be published